# The Abundance of Harmful Rare Homozygous Variants in Children of Consanguineous Parents

**DOI:** 10.3390/biology14030310

**Published:** 2025-03-19

**Authors:** Sankar Subramanian

**Affiliations:** Centre for Bioinnovation, School of Science, Technology, and Engineering, The University of the Sunshine Coast, Moreton Bay, QLD 4502, Australia; ssankara@usc.edu.au; Tel.: +61-7-5430-2873; Fax: +61-7-5430-2881

**Keywords:** consanguineous union, consanguinity, rare variants, autosomal recessive diseases, first cousin, deleterious SNVs

## Abstract

The children born out of marriage between cousins were found to have a higher incidence of recessive genetic diseases than the offspring of unrelated parents. This is because they have many harmful mutations in both of their chromosomes (homozygous). In this study using more than 2500 whole genomes, we show that the children born of the union between double (paternal *and* maternal) first cousins had 20 times more deleterious rare homozygous mutations or single nucleotide variants (SNVs) than those who had unrelated parents. Furthermore, the children of first cousins had 10 times, and the children of second cousins had two times more of these SNVs compared to those present in the offspring of unrelated parents. These results suggest that the children of closely related parents have 20 times higher risk for recessive diseases than those of unrelated parents. These findings have implications for future clinical research. Furthermore, these results could be useful in genetic counseling and public health.

## 1. Introduction

Consanguineous marriage is defined as the union between two individuals related as second cousins or closer [1]. These marriages are still common in many parts of the world, including countries in South Asia, West Asia, the Middle East, and North Africa, involving ~1.1 billion people [2,3]. A number of cultural, social, political, and economic factors influence the people of these regions to prefer marriage between close relatives. A wide range of genetic disorders, including birth malformation, autosomal recessive rare as well as common diseases, were found to be more prevalent in children born of consanguineous union than the offspring of unrelated parents [4,5,6,7,8,9]. The genomes of the former have longer and higher proportions of homozygous segments, such as identical by descent (IBD) and runs of homozygosity (RoH), than those of the latter [1,3,10,11,12,13]. Therefore, the children of consanguineous union are expected to have more harmful rare homozygous variants than the offspring of unrelated parents. This is the major reason for the former to have a higher risk of homozygous recessive genetic diseases and birth malformations than the latter.

A number of clinical studies reported the effect of consanguineous marriages on human health. Previous studies revealed that consanguinity increases the risk of autosomal recessive diseases caused by rare variants [14,15,16,17]. For instance, a large-scale study based on 1929 genes revealed that children of consanguineous couples had a 16.5 times higher risk of autosomal recessive diseases than those of non-consanguineous couples [16]. The whole genome sequences from 6045 Qatar nationals with a high rate of consanguinity revealed that 62.5% of participants carried at least one recessive pathogenic variant [18]. A previous Saudi Arabian study involving 1010 participants found that the children of first cousins were more likely to have cardiovascular diseases, blood diseases, cancer, hearing loss, and speech disorders than those of unrelated parents [19]. Exome sequencing of 2200 genomes from Saudi families identified more than 155 genes to be the candidates for various rare recessive genetic diseases [20]. A study based on 599 Qatari families found a significantly higher risk of autosomal recessive disorder among the children of consanguineous parents than those of unrelated parents [21]. Another study conducted in India found a significantly higher incidence of psychotic disorders, heart disease, hypertension, and cancer in the children of consanguineous parents compared to those of unrelated parents [22]. Furthermore, an earlier study found that autosomal recessive diseases such as neurodegenerative disease (NDD), phenyl keto urea (PKU), and sensorineural deafness (SND) were 2–14 times more common or frequent in the children of consanguineous parents than those of unrelated parents [15]. The rare variants causing congenital disabilities such as intellectual disability, blindness, and physical disability are 3–5 times more prevalent in the children of cousins than those of unrelated parents [23].

Although many studies have shown the association between genetic variants and diseases, the magnitude of the difference in the number of such variants between the consanguineous and unrelated (as control) children is unknown. Hence, it is important to quantify this using empirical genomic data. A previous study calculated the relationship coefficient for >2500 genomes from the 1000 genome project and classified the individuals based on their level of consanguinity [24]. The present study used these data and computed the number of homozygous rare, low-frequency, and common variants in these genomes and estimated the magnitude of difference between those observed for the children of consanguineous and unrelated parents.

## 2. Materials and Methods

### 2.1. Genome Data

The whole genome data for 26 human populations around the globe were obtained from the 1000 genomes project (https://www.internationalgenome.org/data-portal/data-collection/phase-3, accessed on 12 June 2023) [25] including those from Africa (ESN, GWD, LWK, MSL, and YRI), America (ACB, ASW, CLM, MXL, PEL, and PUR), East Asia (CHB, CHS, CDX, JPT, and KHV), Europe (CEU, GBR, FIN, IBS, and TSI), and South Asia (BEB, GIH, ITU, PJL, and STU) (Appendix A). This dataset contained 61 to 113 individuals in each population and a total of 2504 genomes. We defined rare, low-frequency, and common variants as those with DAF < 0.01, 0.01 ≥ DAF < 0.05, and DAF ≥ 0.05, respectively. The term DAF denotes derived allele frequency. The alternate or minor alleles could be either ancestral (wild type) or mutated, and therefore, to identify those mutated in the human lineage, we used one or more ancestors to determine the direction of change (for details, see the data analysis section below). This is important for homozygous SNVs because most of the ancestral alleles are in a homozygous state, and only a small fraction belongs to the mutated alleles. The children of consanguineous parents in these data have been identified by a previous study [24], which found there were 25, 68, 501, and 1910 individuals who were children born out of parents who were double first cousins (father and mother were first cousins through their maternal as well as paternal lineages), first cousins, second cousins, and unrelated persons, respectively (Appendix A).

### 2.2. Data Analysis

To identify the derived homozygous SNVs, the direction of mutational changes was determined by using the ancestral state of the nucleotides. For this purpose, the EPO alignments consisting of six primate genomes were used [26]. This alignment contains the nucleotides of six primates, which provides a high probability of determining the ancestral state of the nucleotide positions on human chromosomes. This method may influence the detection of the ancestral state of some of the variants that were created or mutated after humans separated from chimpanzees (and other ancestors) and substituted throughout the human populations. Therefore, we also used alternate or minor alleles instead of derived alleles to identify deleterious homozygous SNVs. However, the results remained the same because 99.2% of the derived alleles (identified using the EPO alignments) were also minor/alternate alleles. To determine the deleteriousness of an SNV, the combined annotation dependent depletion (CADD) method was used [27]. This method integrates diverse annotations into a single score (C-score). The C-scores were available for the whole genomes (2.86 billion nucleotide positions or sites) and for the sites that were predicted to be deleterious in nature. The per-site SNV estimates were obtained by dividing the number of respective SNVs by their associated sites. We used a threshold of C-score > 20 to define a deleterious SNV. Please note that the term deleterious SNVs used in this paper refers only to those predicted by this method. The total numbers of derived homozygous rare, low frequency, and common SNVs in each genome were calculated, and the average estimates were obtained for the children of parents who were double first cousins, first cousins, second cousins, and unrelated individuals. Similar estimates were obtained for the deleterious SNVs and nSNVs from the genomes. The statistical significance of the difference between the mean estimates was determined by the Z test.

## 3. Results

First, we computed the average number of rare homozygous SNVs for the children of double first cousins, and this number was 1004 (Figure 1A) (Appendix A). This estimate for the children of first cousins was 551, which is roughly 45% smaller than the former. The estimate obtained for the offspring of second cousins was 145, which was 3.8 times smaller than that obtained for the children of first cousins. A similar number was observed for children born of unrelated parents, which was only 75, twice smaller than that of second cousins. The estimates of deleterious rare homozygous SNVs were almost two orders smaller compared to the overall genomic rare homozygous SNVs, and these numbers were 30, 15.5, 3.3, and 1.3 for the children of double first cousins, first cousins, second cousins, and unrelated parents, respectively (Figure 1B). The average numbers of deleterious rare homozygous nonsynonymous SNVs (nSNVs) in the exomes were 18.7, 9.5, 2.1, and 1.0, respectively (Figure 1C) (Appendix A). The differences between the mean estimates obtained for the children of double first cousins, first cousins, second cousins, and unrelated parents were statistically significant using a Z test (*p* < 0.0004).

We then calculated the magnitude of difference between the mean number of SNVs observed for the children of consanguineous parents and those of unrelated parents. Figure 2 shows that the mean genomic rare homozygous SNV value estimated for the double first cousin children was 13.4 times higher than that observed for the offspring of unrelated parents. This difference was even higher—22.8 and 18.9 times for deleterious SNVs and nSNVs, respectively. The children of first cousins had a 7.3 times higher number of genomic rare homozygous SNVs than the children of unrelated parents, and this difference was 11.8 and 9.6 times for the deleterious SNVs and nSNVs, respectively (Figure 2). The children of second cousins had a 1.9 times higher number of genomic rare homozygous SNVs than those of the offspring of unrelated parents. A similar magnitude of the difference was observed for harmful rare homozygous SNVs (2.5 times) and nSNVs (2.2 times).

Finally, analysis of low-frequency and common homozygous SNVs revealed that the mean number of deleterious low-frequency homozygous SNVs of double cousin children was 3.4 times higher than that observed for the offspring of unrelated parents (164 vs. 48) (Figure 3A). This difference was 2.2 times (108 vs. 48) and 1.1 times (55 vs. 48) for the comparisons involving the children of first cousin parents vs. unrelated parents and second cousin parents vs. unrelated parents, respectively (Appendix A). The average count of homozygous common SNVs of the children of unrelated parents (21,293) was only 1.08, 1.05, and 1.01 times (or 8%, 5%, and 1%) higher than that estimated for the offspring of double first (23,099), first (22,334), and second cousins (21,504), respectively (Figure 3B) (Appendix A). The differences between the mean number of SNVs estimated for these (double first vs. first, first vs. second, and second vs. unrelated) were also highly significant (*p* < 0.0006).

We also calculated the proportion of the children of double first/first cousins in different world populations (Figure 4A). This revealed that the populations from South India (STU and ITU) and Pakistan (PJL) had a high proportion (10–30%) of children born from marriage between first cousins. Furthermore, our investigation of world populations revealed that of 26 populations, 17 populations had more than 10% of individuals that were offspring of second cousins (Figure 4B). The populations from Southeast Asia (GIH, ITU, PJL, and STU), Latin American (CLM, PUR, and PEL), and Finnish (FIN) populations had high proportions (>30%) of children of second cousins.

## 4. Discussion

The results of this study predict that the children of consanguineous parents could have 2–20 times more risk of recessive genetic diseases caused by rare variants than the offspring of unrelated parents (Figure 1 and Figure 2). The huge difference could be explained by the elevated proportion of IBD or RoH present in the former [1,3,10,11,12,13]. It is well-known that most of the autosomal recessive diseases are known to be caused by rare variants in homozygous state [1,28]. The union between closely related persons results in rare alleles in homozygous states [28]. This explains the large number of deleterious rare homozygous variants observed in the children of cousins (Figure 1 and Figure 2). The findings of the clinical studies suggest these homozygous variants can potentially be associated with autosomal recessive diseases [14,15,16,23]. The deleterious homozygous SNVs observed in this study included many variants that are known to be associated with recessive diseases. For instance, rare or low-frequency variants causing rare diseases such as hearing impairment [29], corneal dystrophy [30], Stargardt disease [31], epilepsy [32], biotinidase [33], and butyrylcholinesterase deficiencies [34] were identified in South Asian populations (GIH, ITU, PJL, and STU). This study also identified homozygous SNVs associated with oculocutaneous albinism [35], and Kabuki syndrome [36] in African populations. Furthermore, recessive homozygous variants associated with type 2 diabetes [37] and Alagille syndrome [38] in European populations were also identified by this study.

While consanguinity was found to be associated with rare diseases, its impact on complex or multifactorial diseases caused by low frequency or common variants is uncertain [1,14,28]. For instance, a previous study showed no significant association between consanguinity and complex diseases such as diabetes, cancer, epilepsy, and psychiatric disorders typically caused by common variants [23]. Another study showed that the children of consanguineous parents had only a slightly higher risk of diseases such as diabetes mellitus, cancer, mental disorders, asthma, and hypertension [39]. In the present study, we observed that the magnitude of the difference in the number of deleterious homozygous common variants between the children of cousins and those of unrelated parents was small (Figure 3). This is in conformity with the results of earlier clinical studies [1,14,23,28,39]. Some investigators argue that rare variants also could cause common diseases [40]. However, it is unclear whether these variants cause diseases only in the homozygous state. If this is true, then consanguinity might increase the risk of those diseases.

Consanguinity is not common among all populations but is specific to those belonging to certain cultural groups [41]. Typically, marriages between first or double-first cousins are common among Middle Eastern and South Asian (South Indian and Pakistani) populations [19,41,42]. These marriages are preferred in these communities to promote harmonious relations among the members of the family and to keep the wealth within the families [43,44]. These societies also believe that consanguineous marriages are more stable than those between unrelated persons [44,45]. Furthermore, it is believed that marital disputes could be easily solved among cousins if they have known each other for a long time [2,43].

While our study included the populations of South Asians, it did not include populations from the Middle East as those genomes are not publicly available. Therefore, it is important for large-scale genomic studies to make their data publicly available, which will enable other researchers to perform analyses that the original studies missed. Our study also did not include the genomes from Oceania, including Papuans, Australians, Micronesians, and Polynesians, as they have not been sequenced in appreciable numbers. The genomes of the Amish group are also lacking in this study, and inbreeding is known to be common among this group as well [46]. Therefore, large-scale genome sequencing is needed for populations such as the Oceanians and those from other geographic locations that are not represented or underrepresented, as well as populations from specific cultural groups such as the Amish.

## 5. Conclusions

Our results showed that the number of deleterious homozygous variants could be up to 20 times higher in the offspring of cousins compared to those of unrelated parents. This implies that the former have a 20 times higher risk of recessive genetic diseases (caused by rare variants) than the latter. Although the negative effects of consanguineous marriages are known, this result reveals the magnitude of disease risk. This is an important message that could be highlighted to health workers and to the public, which could act as a deterrent of consanguineous marriages. Furthermore, our results suggest that consanguineous marriages are pervasive in most parts of the world, and a significant proportion of individuals in the vast majority of the population around the globe are the children of cousins (Figure 4). This further emphasizes the need to highlight the magnitude of the elevated disease risk resulting from consanguineous marriages. The findings of this study also suggest the importance of pre-marital genetic screening as the causal variants for many recessive genetic diseases (HGMD) are known [47], and thus, we are now able to detect them through genetic testing. Therefore, the results reiterate the need to create awareness about genetic screening through health policies, dialogues with communities, and training programs.

## Figures and Tables

**Figure 1 biology-14-00310-f001:**
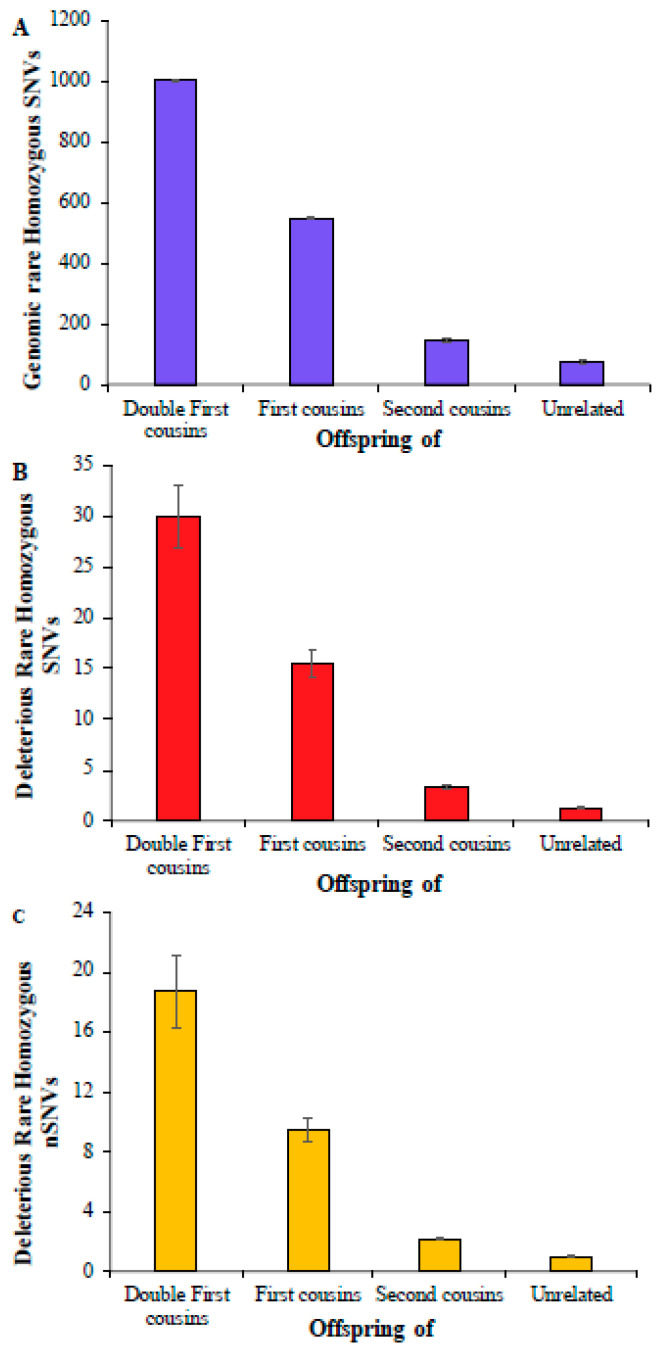
The number of rare homozygous SNVs per genome present in children of consanguineous and unrelated parents. (**A**) Genomic SNVs; (**B**) deleterious SNVs; (**C**) deleterious nonsynonymous SNVs (nSNVs). Error bars denote the standard error of the mean. The differences in the number of SNVs between any two categories (e.g., first vs. second cousins) were statistically significant using the Z-test (*p* < 0.0004).

**Figure 2 biology-14-00310-f002:**
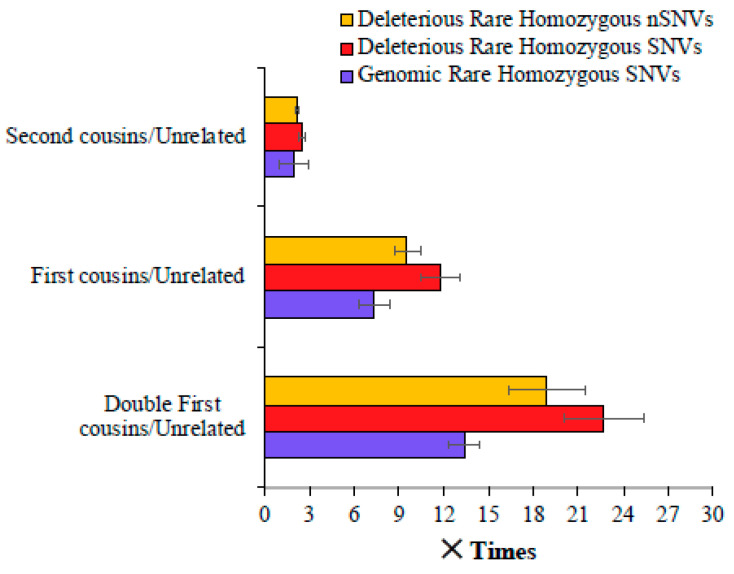
Comparison of genomic and deleterious SNVs in the children of closely related and unrelated parents. To obtain these estimates, the number of SNVs (and nSNVs) present in the children of cousins was divided by those observed in the offspring of unrelated parents. Error bars show the standard error of the mean. The differences in the estimates obtained between any two categories (e.g., first vs. second cousins) were statistically significant using the Z-test (*p* < 0.0007).

**Figure 3 biology-14-00310-f003:**
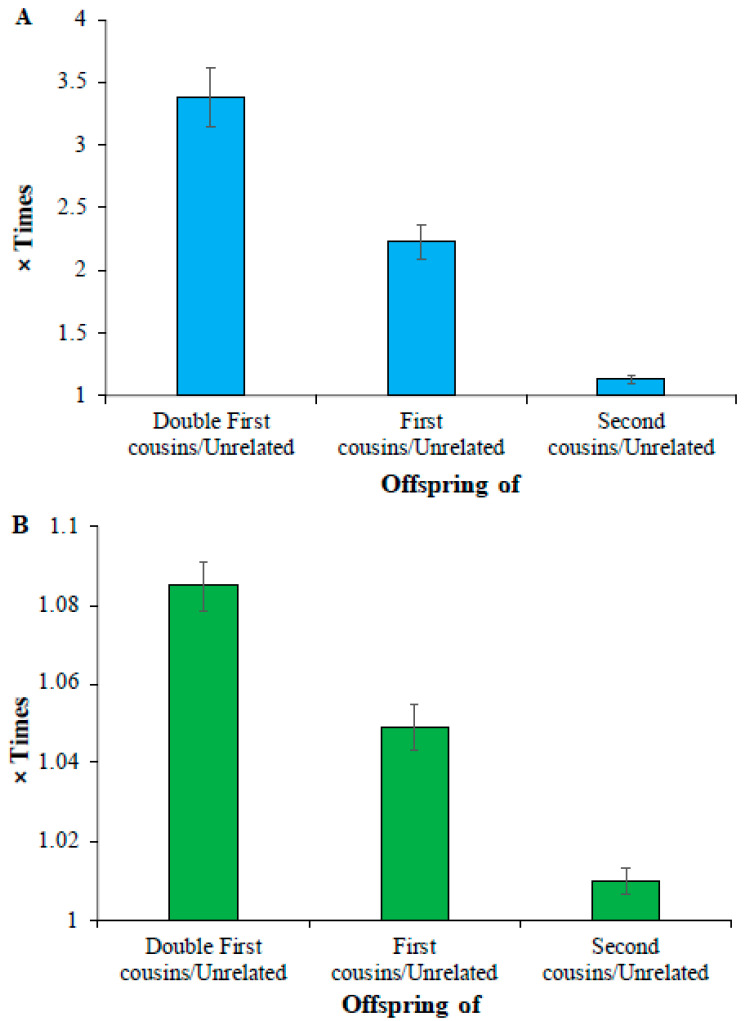
Comparison of deleterious (**A**) low-frequency (0.01 < DAF ≤ 0.05) (**B**) common SNVs (DAF > 0.05) in the offspring of closely related and unrelated parents. The number of deleterious SNVs present in the children of cousins was divided by those observed in the offspring of unrelated parents to obtain these estimates. The error bars indicate the standard error of the mean. The differences in the estimates obtained between any two categories (e.g., double first vs. first cousins) were also highly significant (*p* < 0.0006).

**Figure 4 biology-14-00310-f004:**
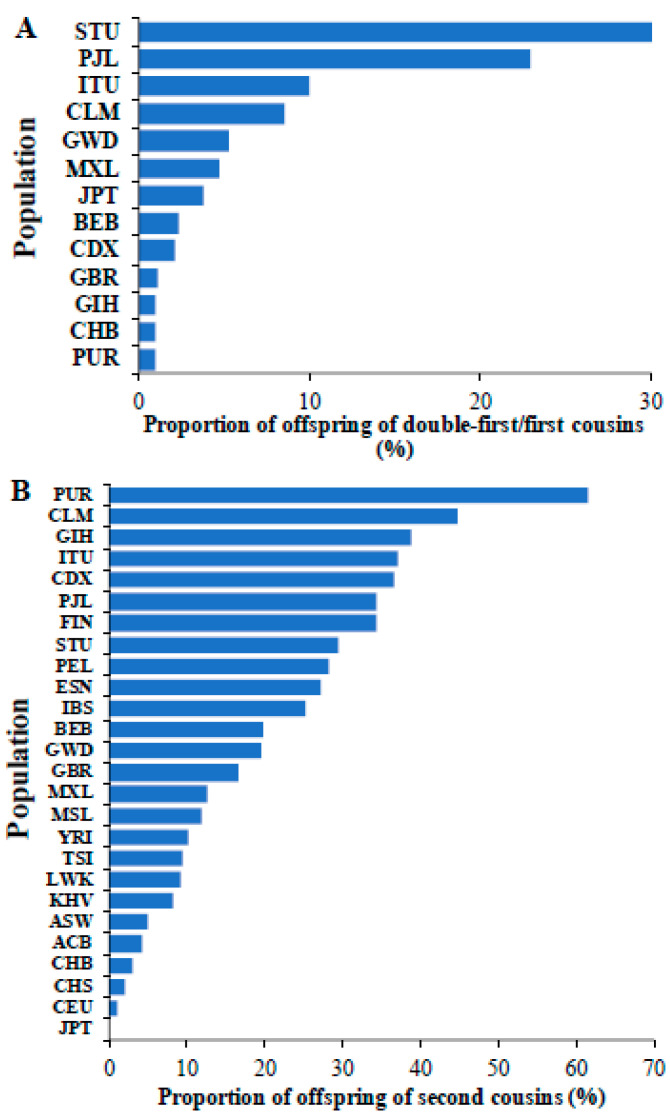
The proportions of children of (**A**) double-first/first cousins and (**B**) second cousins observed in global populations are shown. The populations include those from Africa (ESN, GWD, LWK, MSL, and YRI), America (ACB, ASW, CLM, MXL, PEL, and PUR), East Asia (CHB, CHS, CDX, JPT, and KHV), Europe (CEU, GBR, FIN, IBS, and TSI) and South Asia (BEB, GIH, ITU, PJL, and STU). The populations absent in Figure 4A,B indicate that they have no children born of union between double first/first cousins and second cousins, respectively.

## Data Availability

The genome data and kinship information used in this study are available from Auton et al. (2015) [25] and Gazal et al. (2015) [24], respectively. The estimates generated in the current study are available from the corresponding author on request.

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
