# Peer review of "The Abundance of Harmful Rare Homozygous Variants in Children of Consanguineous Parents"

_biology, 2025, doi:10.3390/biology14030310_

Round 1
Reviewer 1 Report
Comments and Suggestions for Authors
The manuscript "The abundance of harmful homozygous rare variants in children of consanguineous parents" explores the prevalence and impact of harmful homozygous rare variants in children of consanguineous unions, leveraging data from the 1000 Genomes Project. The study provides valuable insights into genetic risks associated with consanguinity, emphasizing public health implications and the need for genetic counseling. While the manuscript is informative, certain areas could be improved for better clarity, depth, and impact.
Major comment:
- Abstract
While concise, the abstract could benefit from more detail on the methodology and specific findings. For instance, the mention of "20 times more deleterious homozygous rare variants" should specify the category (e.g., SNVs) to avoid ambiguity. It will be good if a brief statement included on the implications of the findings for genetic counseling and public health.
- Introduction:
It will be good to include more recent studies on consanguinity’s impact, particularly those focusing on populations not covered by the 1000 Genomes Project.
- Material and Methods:
Data Inclusion: The explanation of the dataset is clear but lacks details on why populations with <90 individuals were excluded, aside from achieving statistical power. Provide more rationale for the choice of thresholds (e.g., DAF ≤0.011 for rare variants).
Ancestral State Determination: The use of EPO alignments for ancestral state determination is mentioned briefly. Expand on this to clarify the reliability and limitations of the approach.
- Results:
Error Bars and Statistical Analysis: While error bars are included in the figures, the manuscript should detail the statistical tests used to compare groups and assess significance.
Population Distribution (Figure 4): The manuscript mentions proportions of consanguineous offspring in different populations but does not analyze potential cultural or historical reasons behind these distributions.
Comparative Analysis: Highlight how these findings align or contrast with other genomic studies of consanguineous populations.
- Discussion:
Address potential biases in the dataset, such as underrepresentation of certain regions or socio-economic groups.
Discuss other possible contributors to the observed differences in variant prevalence, such as environmental factors or selection pressures.
Provide examples of specific autosomal recessive diseases likely influenced by the identified deleterious variants.
Suggest more targeted studies, such as sequencing larger cohorts from underrepresented populations or integrating environmental and epigenetic data.
Minor Suggestions:
Language and Grammar: Minor grammatical corrections and stylistic improvements are needed throughout the manuscript.
For example, line 89: Replace "deleterious homozygous rare variants" with "deleterious rare homozygous variants" for consistency.
Avoid overly technical jargon where simpler terms would suffice.
Author Response
Comment: While concise, the abstract could benefit from more detail on the methodology and specific findings. For instance, the mention of "20 times more deleterious homozygous rare variants" should specify the category (e.g., SNVs) to avoid ambiguity. It will be good if a brief statement included on the implications of the findings for genetic counseling and public health.
Response: The suggested changes have been incorporated in the abstract.
Comment: It will be good to include more recent studies on consanguinity’s impact, particularly those focusing on populations not covered by the 1000 Genomes Project.
Response: As per the suggestion, we have now included the findings of many recent studies (lines 54-75)
Comment: Data Inclusion: The explanation of the dataset is clear but lacks details on why populations with <90 individuals were excluded, aside from achieving statistical power. Provide more rationale for the choice of thresholds (e.g., DAF ≤0.011 for rare variants).
Response: We have now included all genomes and used the standard threshold to define rare, low-frequency, and common variants (lines 93-99). Using the new data and threshold, we have recomputed all estimates reported in the paper.
Comment: Ancestral State Determination: The use of EPO alignments for ancestral state determination is mentioned briefly. Expand on this to clarify the reliability and limitations of the approach.
Response: We have clarified this and also used the minor or alternate alleles instead of derived alleles (using the ancestral states) to identify deleterious rare homozygous SNVs. However, the results remained the same because 99.2% of the derived alleles (identified using the EPO alignments) were also minor/alternate alleles. This was further explained on lines 102-111.
Comment: Error Bars and Statistical Analysis: While error bars are included in the figures, the manuscript should detail the statistical tests used to compare groups and assess significance.
Response: We have used the Z test to determine the statistical significance between different estimates. These were mentioned in methods (lines 124-125) and in the figure legends.
Comment: Population Distribution (Figure 4): The manuscript mentions proportions of consanguineous offspring in different populations but does not analyze potential cultural or historical reasons behind these distributions.
Response: The cultural and historical reasons have now been discussed in the discussion section (lines 203-210).
Comparative Analysis: Highlight how these findings align or contrast with other genomic studies of consanguineous populations.
Response: We have now discussed our results with those of the other studies on consanguineous populations on lines 172-201.
Comment: Provide examples of specific autosomal recessive diseases likely influenced by the identified deleterious variants.
Response: We have now discussed the variants identified in this study and their associations with autosomal recessive diseases (lines 180-188).
Comment: Discuss other possible contributors to the observed differences in variant prevalence, such as environmental factors or selection pressures.
Response: We could not think of any selection pressures or environmental factors influencing our results.
Comment: Address potential biases in the dataset, such as underrepresentation of certain regions or socio-economic groups.
Response: The biases in the dataset and underrepresentation of populations from some regions have been now discussed in lines 211-217.
Comment: Suggest more targeted studies, such as sequencing larger cohorts from underrepresented populations or integrating environmental and epigenetic data.
Response: The suggestions for future targeted studies have been now discussed (lines 217-221).
Comment: Language and Grammar: Minor grammatical corrections and stylistic improvements are needed throughout the manuscript. For example, line 89: Replace "deleterious homozygous rare variants" with "deleterious rare homozygous variants" for consistency. Avoid overly technical jargon where simpler terms would suffice.
Response: Done.
Reviewer 2 Report
Comments and Suggestions for Authors
The manuscript explores the impact of consanguineous unions on the abundance of harmful homozygous rare variants in offspring, with significant implications for public health and genetics research. While the study is well-designed and presents valuable findings, several areas require attention to improve the clarity, accuracy, and presentation of the manuscript:
1. The abstract does not mention the analysis results of low-frequency and common variants, which are discussed in detail in the manuscript (e.g., Figure 3). Including these findings in the abstract would provide a more comprehensive summary of the study’s content and outcomes. Please consider briefly mentioning the results of low-frequency and common variant analysis in the abstract to ensure it accurately reflects the breadth of the study.
2. In the current manuscript, while the axes in the figures (e.g., Figures 1-4) display numeric labels, they lack visible tick marks. I recommend to add tick marks to improve the clarity and scientific rigor of the figures.
3. In the sentence:“ The average count of homozygous common SNVs of the children of unrelated parents was only 9%, 6%, and 0.4% higher than that estimated for the offspring of double first, first, and second cousins, respectively (Figure 3B).”
the percentages (9%, 6%, and 0.4%) mentioned are not clearly visible or directly derivable from Figure 3B. Please verify whether these percentages are accurately represented in Figure 3B. If they are based on calculations not shown in the figure, consider providing clarification in the text or supplementary materials. If these percentages are incorrect or irrelevant, kindly revise the statement for accuracy and consistency with the figure.
4. In the manuscript, there is a typographical error in the phrase "a large-scle study", which appears in the discussion section (line 146 in the PDF). The correct spelling should be "a large-scale study". Please revise this spelling mistake to ensure the manuscript maintains its professional quality and readability.
Author Response
Comment: The abstract does not mention the analysis results of low-frequency and common variants, which are discussed in detail in the manuscript (e.g., Figure 3). Including these findings in the abstract would provide a more comprehensive summary of the study's content and outcomes. Please consider briefly mentioning the results of low-frequency and common variant analysis in the abstract to ensure it accurately reflects the breadth of the study.
Response: The results of low-frequency and common variant analyses have now been included in the abstract.
Comment: In the current manuscript, while the axes in the figures (e.g., Figures 1-4) display numeric labels, they lack visible tick marks. I recommend to add tick marks to improve the clarity and scientific rigor of the figures.
Response: Done
Comment: In the sentence:“ The average count of homozygous common SNVs of the children of unrelated parents was only 9%, 6%, and 0.4% higher than that estimated for the offspring of double first, first, and second cousins, respectively (Figure 3B).” the percentages (9%, 6%, and 0.4%) mentioned are not clearly visible or directly derivable from Figure 3B. Please verify whether these percentages are accurately represented in Figure 3B. If they are based on calculations not shown in the figure, consider providing clarification in the text or supplementary materials. If these percentages are incorrect or irrelevant, kindly revise the statement for accuracy and consistency with the figure.
Response: We have modified these statements to clarify this (lines 157-159).
Comment: In the manuscript, there is a typographical error in the phrase "a large-scle study", which appears in the discussion section (line 146 in the PDF). The correct spelling should be "a large-scale study". Please revise this spelling mistake to ensure the manuscript maintains its professional quality and readability.
Response: Corrected